# Distilling Task-Level Coordination Policies
# for Generalizable Multi-Agent Cooperation

**Zimo Zhai** [1]   **Manjie Xu** [1 2]   **Wei Liang** [1 3]

## Abstract

Large language models have shown strong reasoning abilities and are increasingly explored as high-level coordinators for multi-agent systems. However, directly deploying LLMs for coordination remains challenging, as effective policies often fail to reliably emerge at the low-level control stage, and inference costs limit scalability. We propose **SynCoord** (Synthetic Coordination Distillation), a self-supervised pipeline that distills task-level decision-making for cooperation from high-capacity reasoning models into lightweight agent policies. Our approach does not rely on explicit supervision or handcrafted coordination rules. Instead, we define a set of task-level tool interfaces that constrain LLM interaction and enable the collection of interaction trajectories, which are then used to train compact coordinated policies. This distillation process transfers coordination behaviors that are difficult to elicit through prompting alone, while substantially reducing inference overhead at execution time. We evaluate our method on cooperative multi-agent benchmarks including Overcooked-AI and Level-Based Foraging (LBF), under varying team sizes and environment scales. Experimental results show that the distilled policies achieve success rates and execution efficiency comparable to reinforcement learning–based methods, while exhibiting fewer erroneous or redundant actions. Moreover, the learned task-level coordination policy generalizes effectively to unseen team compositions and larger layouts without retraining.

[1]School of Computer Science & Technology, Beijing Institute of Technology, Beijing, China [2]Institute for AI, Peking University, Beijing, China [3]Yangtze Delta Region Academy of Beijing Institute of Technology, Jiaxing, China. Correspondence to: Wei Liang <liangwei@bit.edu.cn>.

*Proceedings of the 43rd International Conference on Machine Learning*, Seoul, South Korea. PMLR 306, 2026. Copyright 2026 by the author(s).

## 1. Introduction

The development of intelligent agents capable of effective multi-agent collaboration represents a longstanding objective in artificial intelligence(Chen et al., 2023). Many real-world scenarios, ranging from human-robot teaming to multiplayer games, require agents not only to act autonomously but also to coordinate their decisions and behaviors with others in dynamic and partially observable environments(Xu et al., 2023a; DeWeese & Qu, 2024). Successful collaboration in such settings depends critically on the ability to reason about *task-level* dependencies, allocate roles among agents, and adapt coordination strategies as the environment and team composition change.

Recent advances in reinforcement learning (RL) have led to impressive performance in both single-agent tasks and certain multi-agent scenarios(Bauer et al., 2023; Li et al., 2023; Xu et al., 2026a). However, when applied to cooperative environments with diverse layouts, task compositions, or varying team sizes, RL-based approaches often struggle to generalize(Li et al., 2023). These methods typically rely on extensive environment-specific training and implicit coordination emerging from joint optimization, making them sensitive to distribution shifts and difficult to scale. In particular, the absence of explicit task-level coordination mechanisms limits their ability to robustly assign and prioritize tasks across agents under changing conditions.

In contrast, large language models excel at high-level reasoning and structured decision making, and have recently been explored as coordinators in multi-agent systems(Bo et al., 2024). Despite their strengths, directly deploying LLMs as acting agents remains challenging. Effective coordination policies often fail to reliably emerge at the low-level control layer, where temporally extended behaviors must be executed consistently, and the high inference cost of LLMs hinders scalable real-time deployment(Wan et al., 2025). As a result, neither RL-based coordination nor direct LLM-based control alone provides a satisfactory solution for robust and efficient multi-agent collaboration. We argue that effective multi-agent cooperation should be modeled as a transferable task-level coordination abstraction, rather than emerging implicitly from low-level control optimization.

To address this challenge, we propose **SynCoord**, a self-supervised distillation pipeline that utilizes self-supervised distillation to translate the latent reasoning of large language models into emergent cooperative behaviors. With the rapid development of Large Language Models (LLMs) technology, two distinct trends have emerged: compact models with constrained parameters that enable efficient real-time decision making, and large-scale models that, while computationally intensive, exhibit strong capabilities in structured reasoning and instruction following(Bai et al., 2023; Touvron et al., 2023; Agarwal et al., 2025; Liu et al., 2024; Team et al., 2023). Building on these observations, we introduce a two-stage distillation pipeline for multi-agent coordination, in which large language models are used to generate coordination supervision rather than to directly act in the environment. Specifically, a lightweight Intuitive Agent first produces preliminary action suggestions based on the current environment state, providing fast and low-cost guidance. Conditioned on these executable behaviors, two high-capacity Thinking Agents engage in cooperative reasoning to generate structured interaction trajectories that resolve inter-agent dependencies and coordination conflicts. Finally, a Distilled Agent is trained on the interaction data produced by the Thinking Agents, learning to execute efficient and robust two-agent coordinated behaviors without requiring large-scale model inference at runtime.

The intuition behind our design of SynCoord stems from the observation that while task execution can be learned through local experience, sophisticated multi-agent cooperation is an emergent capability that manifests at the scale of high-capacity reasoning models(Zhang et al., 2024; Tran et al., 2025; Zhu et al., 2025). The Intuitive Agent establishes the basic environmental competence, providing a grounded behavioral substrate upon which the Thinking Agent orchestrates collaborative strategies. This design allows the latent cooperative priors in large-scale models—such as social reasoning and recursive theory-of-mind—to emerge and be captured as structured coordination trajectories. By distilling these emergent patterns into a lightweight policy, we bridge the gap between heavy-duty deliberative intelligence and real-time execution. The resulting Distilled Agent does not merely replicate actions; it inherits the systematic coordination logic of the larger model, enabling it to maintain high-level social intelligence and robust generalization across dynamic settings while operating within the latency constraints of real-time deployment.

We evaluate our method on the cooperative benchmark Overcooked-AI(Carroll et al., 2019), where agents must coordinate in real time under partial observability to complete complex recipes. Results show that the Distilled Agent achieves high success rates and efficient execution comparable to reinforcement learning–based methods, while exhibiting fewer erroneous or redundant actions. Moreover, the

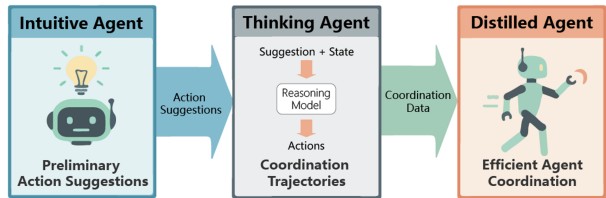

*Figure 1.* **Overview of the proposed task-level coordination under self-generated supervision and distillation framework.** We separate high-level multi-agent reasoning from low-level execution, using large language models to generate coordination trajectories that are distilled into lightweight policies for efficient deployment.

learned coordination policy generalizes across diverse layouts, unseen team compositions, and larger spatial environments without retraining, demonstrating strong scalability and transferability across cooperative settings.

In summary, our contributions are threefold. (1) We introduce a two-stage distillation pipeline for scalable multi-agent coordination, in which lightweight and large-scale language models play distinct roles in supervision generation and deployment, rather than acting jointly in the environment. (2) We propose a self-supervised data collection and distillation framework, where coordination trajectories are generated through cooperative reasoning between high-capacity language models and distilled into a compact agent capable of efficient execution. (3) We demonstrate that the learned task-level coordination policy generalizes across diverse layouts, unseen task compositions, team sizes, and larger spatial environments, achieving scalable cooperation comparable to RL-based methods while avoiding the inference overhead of direct LLM deployment.

## 2. Related Work

### 2.1. Cooperative Agents

Cooperative intelligence concerns scenarios where multiple agents work jointly toward a shared goal(Jin et al., 2025), aiming either to improve efficiency or to tackle tasks inherently requiring joint effort. Prior studies have highlighted key challenges in multi-agent cooperation, including handling non-stationarity(Hernandez-Leal et al., 2017; Papoudakis et al., 2019), ensuring scalability(Christianos et al., 2021), and effectively sharing information during both training and execution(Jiang & Lu, 2018; Sun et al., 2024). While traditional MARL methods have achieved strong performance in fixed cooperative settings, their coordination behaviors are often tightly coupled to specific environment distributions and team configurations, limiting transferability under unseen layouts, task compositions, or larger agent populations(Xu et al., 2023b; Ning & Xie, 2024; Jin et al., 2025). Recent works therefore explore whether high-level

reasoning from large language models can provide more flexible coordination strategies(Chen et al., 2023; Guo et al., 2024). In contrast to approaches that directly use LLMs as acting agents, our work focuses on distilling transferable task-level coordination abstractions into lightweight executable policies.

## 2.2. LLM for Agent Planning

In many studies, large language models (LLMs) have been adopted as high-level planners for agents due to their broad world knowledge and few-shot adaptability(Ahn et al., 2022; Liu et al., 2023). A common paradigm involves decomposing semantic goals into subgoals, calling structured tools for execution, and integrating feedback through memory or reflection mechanisms(Huang et al., 2023; Yao et al., 2022; Huang et al., 2024; Zhai et al., 2026). For instance, Song et al. (2023) apply LLMs to embodied decision-making, while other approaches study active task disambiguation(Kobalczyk et al., 2025), and TinyAgent further explores efficient deployment under resource constraints(Erdogan et al., 2024). In multi-agent settings, LLMs enable abstract cooperation and cross-task adaptation, showing promise for error recovery and long-horizon planning(Nayak et al., 2024). Our method integrates LLM planners with structured function calling to bridge symbolic plans with executable actions(Lin et al., 2025). Framing environment dynamics and agent actions as executable code or structured tools has also been shown to help LLMs overcome semantic inertia and adhere strictly to in-context logical constraints rather than pre-trained priors (Xu et al., 2026b). However, existing systems often lack iterative learning and robust environmental feedback, limiting sustained coordination abilities(Ning & Xie, 2024).

## 2.3. Knowledge Distillation and Synthetic Supervision

Knowledge distillation aims to transfer the capabilities of a large "teacher" model into a smaller "student" model(Mirzadeh et al., 2020). In the context of LLMs, this has evolved into using high-capacity models to generate synthetic reasoning chains or trajectories to train smaller models(Yang et al., 2025; Tian et al., 2025). While traditional "Self-Play" in RL involves agents improving through iterative competition(Silver et al., 2017; Vinyals et al., 2019), recent "LLM Self-Correction" or "Multi-Agent Debate" frameworks use LLM interactions to refine reasoning quality(Pan et al., 2024; He et al., 2025; Zhang et al., 2025).

Our work departs from traditional self-play by employing a two-stage distillation pipeline. Instead of agents playing to maximize a reward signal, our "Thinking Agents" act as a deliberative supervisor, refining the "Intuitive Agent's" suggestions into coordinated trajectories. This process distills coordination and conflict-resolution capabilities of large-scale models into a compact "Distilled Agent." Unlike standard reasoning distillation approaches focusing on improving single-agent reasoning quality or final-answer accuracy, our framework distills structured coordination behaviors emerging from multi-agent deliberation. The student model learns not only executable actions, but also transferable coordination patterns for task allocation and conflict resolution.

# 3. The SynCoord Framework

We present our self-supervised coordination pipeline for multi-agent collaboration, which leverages the complementary strengths of deliberative reasoning and efficient execution. We begin by formally defining the multi-agent collaborative task and then detail our two-stage pipeline. To simplify inter-agent communication and enable a clear division of responsibilities, we assume that a task, once allocated to one agent, is not reassigned to the other agent.

## 3.1. Task Definition

We consider a partially observable multi-agent cooperative environment, formalized by the tuple $\mathcal{M} = \langle \mathcal{N}, \mathcal{S}, \{\mathcal{T}_i\}_{i \in \mathcal{N}}, \{\mathcal{O}_i\}_{i \in \mathcal{N}}, T, \Omega \rangle$. Here, $\mathcal{N} = \{1, \ldots, n\}$ is the set of agents; $\mathcal{S}$ is the global state space; $\mathcal{O}_i$ is the local observation space for agent $i$, and $\Omega : \mathcal{S} \to \prod_{i \in \mathcal{N}} \mathcal{O}_i$ is the observation function. Unlike standard step-wise control, we define $\mathcal{T}_i$ as a *semantic task space* (e.g., executable function calls) rather than a low-level action space, with $T$ representing the environment transition after task execution.

Whenever task reassignment is triggered, agent $i$ receives a local observation $o_i^t = \Omega_i(s^t)$ and selects a semantic task $\tau_i^t \in \mathcal{T}_i$ based on its policy $\pi_i$. The joint task assignment $\boldsymbol{\tau}^t = (\tau_1^t, \ldots, \tau_n^t)$ is then executed in the environment, leading to a new state $s^{t+1}$. Instead of maximizing a step-wise numerical reward, the objective is to learn a set of task-level policies $\{\pi_i\}_{i \in \mathcal{N}}$ that coordinate to accomplish a shared global goal (e.g., completing all required recipes) within minimal execution timesteps.

## 3.2. Overview

Our framework adopts a two-stage design that explicitly separates deliberative coordination from efficient execution. The core idea is to use a high-capacity language model to generate task-level coordination supervision, which is then distilled into lightweight policies suitable for real-time multi-agent control under strict latency constraints.

**Thinking Agent.** The Thinking Agent is a large language model with access to the global environment state and full multi-agent context. It performs deliberative multi-agent reasoning to resolve task dependencies and coordination conflicts, producing task assignments together with structured coordination rationales that explain the underlying

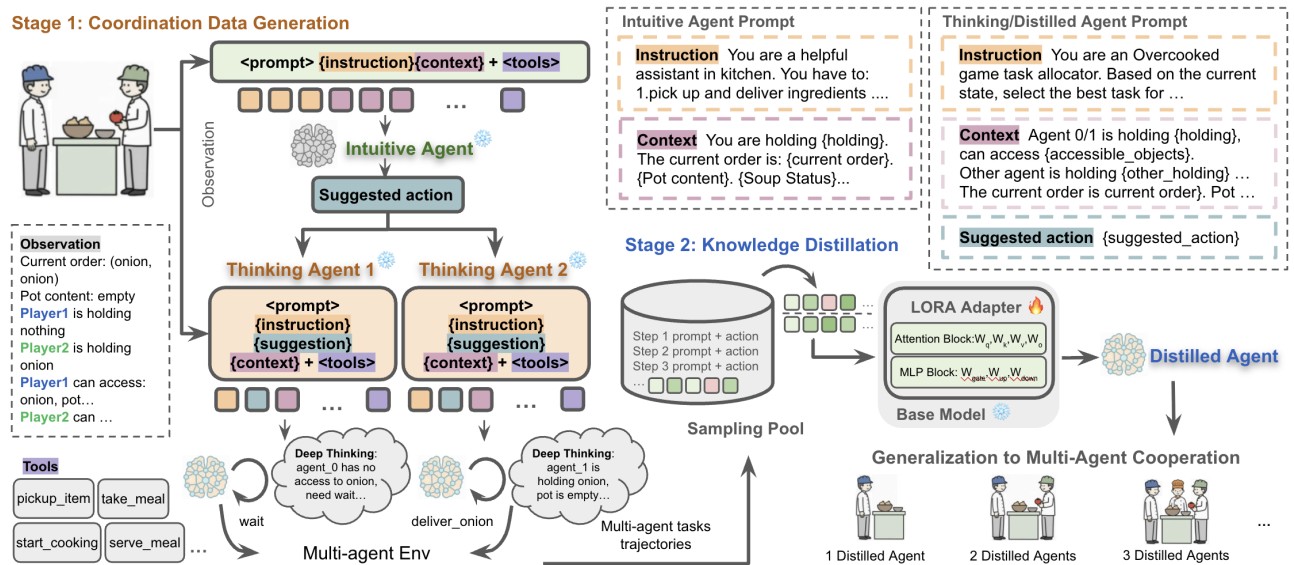

*Figure 2.* **Two-stage coordination pipeline with reasoning-to-execution distillation.** A high-capacity Thinking Agent performs global reasoning and coordination, while the distilled agent executes task-level decisions efficiently using local observations without thinking.

decisions. Importantly, the Thinking Agent is used only during offline data generation and does not act directly at deployment time, avoiding runtime computational overhead.

Given the current state, the Thinking Agent outputs a tuple $(\tau^t, e^t, c^t)$, where $\tau^t$ denotes semantic task assignments, $e^t$ contains agent-specific coordination explanations grounded in local roles, and $c^t$ summarizes the global coordination rationale. The reasoning outputs $(e^t, c^t)$ are optional. Each assigned task is executed through a corresponding sequence of low-level environment actions until task completion or significant environmental changes occur.

**Distilled Agent.** The Distilled Agent represents the execution-time policy. It operates under partial observability using only locally available execution-time information, and is implemented as a compact language model that predicts executable function calls with minimal inference cost. The Distilled Agent does not have access to privileged global information, such as the full environment state or coordination rationales.

During training, the distilled agent is supervised using only local observations and the corresponding task assignments generated by the Thinking Agent, preventing information leakage from privileged global states.

The Thinking Agent generates a coordination dataset

$$\mathcal{D}_{\text{coop}} = \{(s^t, \tau^t)\}_{t=0}^{T}$$

where $\tau^t$ denotes task-level assignments, and $\mathbf{a}_\tau^t$ denotes the corresponding low-level action trajectories induced during execution.

Optional coordination annotations, such as explanations or

global rationales, may be generated during rollout collection to improve coordination consistency.

The distilled policy is trained only on local observations together with task-level assignments $\tau^t$. Unlike reactive policies that repeatedly replan low-level actions, SynCoord maintains persistent task assignments until task completion or significant environmental changes occur.

All privileged information, including global state access and coordination rationales, is discarded before training the execution-time policies to ensure fair, realistic evaluation.

### 3.3. Preliminary Intuitive Agent Bootstrap

To mitigate the cold-start issue during coordination data generation, we introduce a preliminary bootstrap phase. Since the Thinking Agent is instantiated from a general-purpose foundation model without task-specific planning priors, its initial coordination outputs may be invalid or inefficient in complex environments.

We therefore train a lightweight bootstrap Intuitive Agent $\pi_{\text{bootstrap}}$ to capture basic environment dynamics and individual agent behaviors. This policy is trained via supervised fine-tuning on single-agent trajectories:

$$\mathcal{L}_{\text{bootstrap}}(\phi) = \mathbb{E}_{(o^t, \tau^t) \sim \mathcal{D}_{\text{single}}} \left[ -\log \pi_{\text{bootstrap}}(\tau^t \mid o^t; \phi) \right]. \tag{1}$$

During data collection, $\pi_{\text{bootstrap}}$ is used only to provide executable task proposals that guide the Thinking Agent toward valid and environment-consistent coordination plans during early exploration. It does not act as a deployment policy and does not override the Thinking Agent's decisions,

serving purely as a facilitation mechanism.

### 3.4. Knowledge Distillation

To obtain executable policies under partial observability, we decompose the Thinking Agent's trajectories into local observation-task training data. We construct a shared distillation dataset $\mathcal{D}_{\text{dist}} = \{(o_i^t, f_i^t)\}_{i,t}$, where $o_i^t = \Omega_i(s^t)$ denotes the local observation available at execution time, and $f_i^t$ is the function call corresponding to the task assignment selected by the Thinking Agent.

The Distilled Agent is implemented as a compact language model that maps local observations to function calls:

$$f_i^t \sim \pi^{\text{Distilled}}(\cdot \mid o_i^t; \theta), \tag{2}$$

where the predicted function call is executed through the environment interface:

$$a_i^t = \text{execute}(f_i^t). \tag{3}$$

The Distilled Agent is trained via behavior cloning using a standard cross-entropy loss over function calls for supervised distillation. All privileged information, including global state access and coordination rationales, is excluded during training and execution, ensuring consistency between training and deployment conditions.

### 3.5. Pipeline Integration

---

**Algorithm 1** SynCoord: Task-Level Coordination

---

1: **Phase 1: Bootstrap (Intuitive Agent Pretraining)**
2: Collect single-agent trajectories $\mathcal{D}_{\text{single}}$
3: Train *Intuitive Agent* $\pi_{\text{int}}$ to execute basic tasks via SFT on $\mathcal{D}_{\text{single}}$
4: **Phase 2: Data Generation (Thinking Agent Supervision)**
5: **for** episode $k = 1$ to $K$ **do**
6:     Initialize environment state $s^0$
7:     **while** task not completed **do**
8:         Obtain task proposals from $\pi_{\text{int}}$ for current state $s^t$
9:         *Thinking Agent* $\pi_{\text{think}}$ reasons over $s^t$ and proposals:
10:             Generate coordinated task assignments $\tau^t = (\tau_1^t, \ldots, \tau_n^t)$
11:             (Optional) Generate coordination rationales $\mathbf{e}^t$
12:         Agents execute tasks $\tau^t$ until completion or state change
13:         Store global transition $(s^t, \tau^t)$ in $\mathcal{D}_{\text{coop}}$
14:     **end while**
15: **end for**
16: **Phase 3: Distillation (Distilled Agent Training)**
17: **for** each agent $i \in \{1, \ldots, n\}$ **do**
18:     Map global states $s^t \in \mathcal{D}_{\text{coop}}$ to local observations $o_i^t = \Omega_i(s^t)$
19:     Add $(o_i^t, f_i^t)$ to $\mathcal{D}_{\text{dist}}$
20: **end for**
21: Train *Distilled Agent* $\pi^{\text{dist}}$ via behavior cloning on $\mathcal{D}_{\text{dist}}$
22: **Deployment**
23: Execute $\pi^{\text{dist}}$ in real-time using only local observations $o_i^t$

---

## 4. Experiments

### 4.1. Experimental Setup

We evaluate the SynCoord framework in the Overcooked-AI environment(Carroll et al., 2019), a benchmark requiring real-time multi-agent coordination under partial observability and dynamic task sequencing. To further assess cross-environment generalization, we additionally evaluate SynCoord on the Level-Based Foraging (LBF) benchmark(Papoudakis et al., 2020), where agents must coordinate to collect items with varying levels. We employ Qwen2.5-7B as the base model for both the Intuitive and Distilled Agents to ensure a balance between reasoning performance and execution speed. The Thinking Agent, which provides high-level coordination supervision, is powered by the official Qwen3-235B-A22B model to leverage its superior deliberative capabilities. For the distillation process, we curate a dataset of approximately 1,200 high-quality interaction trajectories generated by the Thinking Agent. These trajectories contain semantic task-level coordination assignments together with their corresponding executable action trajectories, which are distilled into the 7B model using supervised fine-tuning. To bridge high-level planning with environmental execution, our framework utilizes a structured tool interface—abstracting low-level navigation into semantic tasks to enable the agents to prioritize strategic cooperation and task dependency resolution over grid-level movement.

**Layouts.** We evaluate our method across five standard Overcooked-AI layouts(Carroll et al., 2019), which present a diverse spectrum of coordination challenges. These benchmarks are characterized by varying degrees of functional overlap and spatial constraints, ranging from *Disjoint* layouts that necessitate forced cooperation across separated regions, to *Open* and *Ring* configurations (e.g., Cramped Room and Counter Circuit) where high spatial interference and potential collisions demand precise navigation. We also include *Overlap* layouts to test the policy's ability to handle asymmetric advantages and resource redundancy. To ensure robustness, each layout comprises 13 distinct scene instantiations with randomized ingredient placements, resulting in a total training distribution of $5 \times 13$ scenarios. Furthermore, we extend the original two-agent configurations to accommodate teams of up to three agents, allowing us to assess the scalability of our coordination policy under increased task complexity.

**Recipe.** Recipes differ in the number and type of ingredients and required cooking steps. We include both 13 *seen* recipes (used during single agent and SynCoord training) and 17 *unseen* recipes (not encountered during training) to evaluate the compositional generalization of our model.

**LBF Benchmark.** We additionally evaluate SynCoord on

the Level-Based Foraging (LBF) benchmark(Papoudakis et al., 2020) using the official EPYMARL setup. To evaluate scalability and transferability, SynCoord is trained only on 1-agent and 2-agent environments and evaluated zero-shot on unseen 3-agent configurations and larger layouts without retraining.

**Evaluation metrics.** We report three key metrics to evaluate coordination performance: **Success Rate (SR)**, the fraction of episodes where all recipes are completed; **Average Success Timesteps (Avg T)**, the total steps required for completion, reflecting execution efficiency; and **Average Success Shaped Reward (Avg R)**, which captures cumulative intermediate progress and interaction precision. Importantly, a higher Avg R does not necessarily indicate better performance, as it can be inflated by redundant or non-essential object interactions; therefore, Avg R should be interpreted jointly with success rate and execution efficiency when evaluating coordination quality.

To comprehensively evaluate the proposed SynCoord framework, our experiments are designed around three complementary aspects. First, we perform a *main performance comparison* between the distilled SynCoord model and both RL-based and LLM-based baselines, measuring success rate, average timesteps, and shaped rewards across all canonical layouts and reporting both per-layout and aggregated metrics. Second, we assess *generalization capabilities* of the SynCoord along two dimensions: recipe generalization, by evaluating on both seen and previously unseen recipes, and team-size generalization, by comparing independent single-agent execution with two-agent and three-agent cooperative scenarios. Finally, we conduct a set of *ablation studies* to quantify the contributions of each design choice. All experiments are averaged over 5 seeds to account for stochasticity in both environment dynamics and policy initialization.

### 4.2. Main Results on Overcooked-AI

We evaluate our task-level coordination policy, **SynCoord**, against two categories of baselines: RL-based multi-agent methods, including MAPPO, RMAPPO(Yu et al., 2022), and IPPO(De Witt et al., 2020), and prompt-based LLM policies, including GPT-5.2(Agarwal et al., 2025), Gemini-3-flash(Team et al., 2023), and Qwen3(Bai et al., 2023). For LLM baselines, we maintain the same prompt structure as used in SynCoord to ensure a fair comparison of reasoning capabilities. Regarding the RL baselines, we restrict their evaluation to simpler distributions ($1 \times 1$ and $1 \times 13$) because standard MARL methods struggle with high optimization complexity and often fail to converge or exhibit unstable behavior when trained on the full, diverse $5 \times 13$ layout distribution.

As shown in Table 1, a key distinction lies in the training distribution. While our SynCoord is trained once on a broad

distribution covering 5 layouts with 13 scenes each ($5 \times 13$), RL baselines struggle significantly with generalization. When the distribution expands from a single scene ($1 \times 1$) to $1 \times 13$, the success rate (SR) of RL methods collapses to approximately 20%. Furthermore, RL methods fail to converge or yield highly unstable behavior on the full $5 \times 13$ training set. Consequently, the RL results in Table 1 represent their performance on the simplified layouts where they could be reliably trained.

**Comparison with RL Baselines.** On the subset of layouts where RL baselines are viable, SynCoord achieves a comparable success rate of 97.4%. Notably, SynCoord maintains a lower average completion time (Avg T = 39.97) than most RL baselines, indicating that our policy's execution is at least as efficient as specialized RL coordination. An important observation is that SynCoord yields a lower average shaped reward (Avg R) than MAPPO (35.12 vs. 39.77). In this environment, the shaped reward reflects the frequency of object interactions, where higher reward often indicates redundant or non-essential actions that inflate interaction counts. The lower reward of SynCoord, coupled with its high success rate, suggests a more precise and conservative execution, achieving task completion with fewer unnecessary or misaligned actions.

**Comparison with LLM Baselines.** When compared against training-free LLM baselines on the full $5 \times 13$ distribution, SynCoord demonstrates a superior balance between reasoning and execution efficiency. While state-of-the-art LLMs like Qwen3-235B and Gemini-3-flash achieve impressive success rates (98% and 96% respectively), they exhibit significantly higher average completion timesteps compared to SynCoord (39.97). This disparity suggests that while LLMs possess strong zero-shot reasoning for high-level task allocation, they lack the temporal consistency and execution precision required for optimal multi-agent coordination, often leading to "trial-and-error" steps.

*Table 1.* Performance comparison under different training distributions. SynCoord and LLM agents are trained or tested on $5 \times 13$ layouts, while RL baselines are trained on a single layout or $1 \times 13$ layouts. All methods are evaluated on their respective training layouts.

| Method | Train/Test | SR ↑ | Avg T ↓ | Avg R — |
|---|---|---|---|---|
| MAPPO | $1 \times 1$ | 1.00 | 42.44 | 39.77 |
| MAPPO | $1 \times 13$ | 0.21 | 43.63 | 49.00 |
| RMAPPO | $1 \times 1$ | 1.00 | 38.49 | 37.26 |
| RMAPPO | $1 \times 13$ | 0.18 | 39.14 | 36.14 |
| IPPO | $1 \times 1$ | 1.00 | 38.49 | 37.26 |
| IPPO | $1 \times 13$ | 0.21 | 65.13 | 46.49 |
| GPT-5.2 | $5 \times 13$ | 0.86 | 48.25 | 36.28 |
| Gemini-3-flash | $5 \times 13$ | 0.96 | 44.24 | 35.25 |
| Qwen3-235B-A22B | $5 \times 13$ | 0.98 | 46.88 | 35.28 |
| **SynCoord** | $5 \times 13$ | 0.97 | **39.97** | **35.12** |

## 4.3. Generalization to Unseen Task Compositions

To measure zero-shot generalization, we evaluate each method on unseen recipes that never appear in single-agent pretraining or two-agent data collection.

We evaluate generalization using both success rate and execution timesteps. While success rate reflects whether agents can complete unseen recipes, timesteps capture the efficiency and coordination quality of the resulting behaviors. Reporting both metrics is crucial, as success alone does not reveal degraded coordination under distribution shifts. Results suggest that task-level coordination facilitates compositional generalization across unseen recipe combinations.

*Table 2.* Generalization to 17 unseen recipes across all layouts.

| Layout | Seen SR | Unseen SR | Drop |
|---|---|---|---|
| Disjoint | 1.00 | 0.99 | 0.01 |
| Overlap | 1.00 | 1.00 | 0.0000 |
| Open | 0.92 | 0.87 | 0.05 |
| Ring1 | 1.00 | 0.98 | 0.02 |
| Ring2 | 1.00 | 0.97 | 0.03 |
| Avg | 0.98 | 0.97 | 0.02 |

## 4.4. Scalability with Team Size

To evaluate whether the learned coordination abstraction transfers across different team sizes, we test SynCoord under random layouts while varying the number of simultaneously controlled agents from one to three. As shown in Fig. 3, the first column reports the *Intuitive Agent* baseline, which is trained to act in the environment but *not* for multi-agent coordination. The remaining columns report SynCoord distilled agents when coordinating 1–3 agents.

Overall, the performance remains strong as team size increases. The SynCoord distilled agent achieves near-perfect success rates when coordinating one or two agents across all random scenarios, and maintains a high overall success rate of 82.8% with three agents. This suggests that the learned coordination abstraction can generalize to larger team sizes without retraining or team-size-specific tuning.

As expected, the average timesteps to completion increase moderately with larger teams, reflecting additional coordination overhead rather than a degradation in decision quality. In several scenarios, adding agents reduces completion time relative to the Intuitive Agent baseline, indicating effective parallelization enabled by task-level coordination.

We focus on SynCoord in this analysis, as reliably training baseline MARL methods under these larger multi-agent settings proved challenging in our experiments. Although the system demonstrates strong generalization, success rates decline in certain three-agent tests. Detailed inspection of

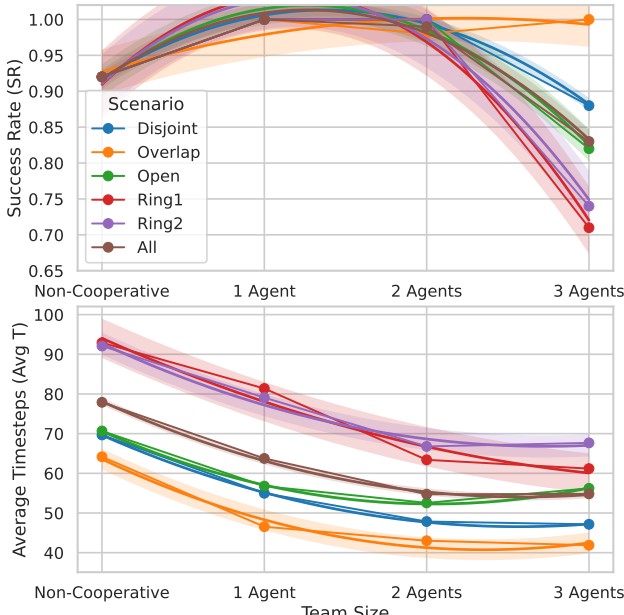

*Figure 3.* **Performance of the task-level coordination policy under different team sizes.** We report Success Rate (SR) and Average Success Timesteps (Avg T) in different settings. The shade represents the standard deviation of the residuals around the fitted polynomial curve, illustrating an approximate range of variability but not a true statistical confidence interval.

these failure modes confirms that the bottleneck is not coordination logic, but severe spatial crowding. In constrained maps, the addition of a third agent leads to significant contention for physical space, causing frequent collisions and blocking behavior. Because our current task abstraction does not explicitly account for agent-to-area density, high-level plans often fail at the execution level due to physical deadlock. This underscores the challenge of managing coordination in high-density environments where spatial interference becomes a dominant factor.

## 4.5. Cross-Environment Generalization on LBF

To evaluate whether task-level coordination generalizes beyond Overcooked-style collaboration, we conduct additional experiments on the Level-Based Foraging (LBF) benchmark, where agents must coordinate under sparse rewards, larger spatial layouts, and dynamically varying cooperative requirements.

To rigorously evaluate zero-shot scalability, SynCoord is trained exclusively on 1- and 2-agent scenarios (e.g., 8x8-2p-2f-coop), while all 3-agent environments remain entirely unseen during training. We evaluate the learned coordination policy directly on larger and more complex layouts, including 10x10-3p-3f and 15x15-3p-5f configurations, without any finetuning or team-size-specific adaptation.

We compare SynCoord against standard MARL baselines,

*Table 3.* Zero-shot generalization on the Level-Based Foraging (LBF) benchmark. SynCoord is trained only on 1-agent and 2-agent environments and evaluated directly on unseen 3-agent settings and larger layouts without retraining.

| Environment | Method | SR ↑ | Avg T ↓ | Avg R ↑ |
|---|---|---|---|---|
| 8×8-2p-2f-coop | MAPPO | 1.00 | 13.00 | 1.00 |
| | IPPO | **1.00** | **11.20** | **1.00** |
| | **SynCoord** | 1.00 | 16.20 | 1.00 |
| 10×10-3p-3f | MAPPO | 1.00 | **22.40** | 1.00 |
| | IPPO | 1.00 | 28.20 | 1.00 |
| | **SynCoord** | 1.00 | 23.40 | 1.00 |
| 15×15-3p-5f (50 steps) | MAPPO | 0.00 | 50.00 | 0.38 |
| | IPPO | 0.00 | 50.00 | 0.50 |
| | **SynCoord** | **0.40** | **42.20** | **0.74** |
| 15×15-3p-5f (100 steps) | MAPPO | 0.00 | 100.00 | 0.45 |
| | IPPO | 0.00 | 100.00 | 0.64 |
| | **SynCoord** | **0.80** | **54.80** | **0.85** |

*Table 4.* Ablation study on bootstrap supervision, teacher capacity, and intuitive agent capacity. **SR**: Success Rate, **Avg T**: Average Success Timesteps, **Data Acc**: Teacher Data Collection Accuracy.

| Ablation | Setting | SR ↑ | Avg T ↓ | Data Acc ↑ |
|---|---|---|---|---|
| Bootstrap | SynCoord | **0.95** | 46.65 | **0.97** |
| | w/o Bootstrap | 0.65 | 45.87 | 0.90 |
| Teacher Capacity | Qwen3-30B | 0.92 | 46.92 | 0.945 |
| | Qwen3-8B | 0.80 | 46.51 | 0.69 |
| Student Capacity | 7B Intuitive Agent | **0.95** | 46.65 | **0.97** |
| | 1.5B Intuitive Agent | 0.00 | – | 0.92 |

including MAPPO and IPPO, implemented using the official EPYMARL framework with 5 random seeds. In LBF, higher shaped rewards directly correlate with successful cooperative collection efficiency.

As shown in Table 3, SynCoord demonstrates strong zero-shot scalability across both team size and environment scale. Although trained only on two-agent data, SynCoord generalizes perfectly to the unseen 10x10-3p-3f setting, achieving a 1.00 success rate without any team-specific adaptation. This suggests that the learned task-level coordination abstraction transfers naturally across different agent compositions.

More importantly, in the substantially harder 15x15-3p-5f environment, both MAPPO and IPPO struggle to solve the task even with extended horizons, while SynCoord maintains a 40% success rate within 50 steps and reaches 80% success within 100 steps.

We observe that RL baselines tend to exhibit locally reactive behaviors that struggle under large-scale sparse-reward coordination, leading to inefficient exploration and poor subtask allocation. In contrast, SynCoord explicitly reasons over semantic task assignments, enabling agents to avoid conflicting objectives and distribute cooperative subtasks more effectively. These results suggest that task-level coordination provides strong inductive priors for scalable multi-agent cooperation, particularly under unseen team sizes and expanded spatial layouts.

### 4.6. Ablation Studies

Table 4 presents a comprehensive ablation analysis of SynCoord, focusing on three key components: (i) the bootstrap phase based on executable single-agent behaviors, (ii) the capacity of the teacher reasoning model, and (iii) the capacity of the distilled intuitive coordination agent.

First, we evaluate the importance of the bootstrap phase. Removing the bootstrap stage significantly reduces the success rate from 0.95 to 0.65. This result demonstrates that executable single-agent priors are critical for overcoming the cold-start problem in cooperative environments. Without such initialization, the reasoning teacher frequently generates unexecutable actions during early exploration due to physical constraints and sparse coordination signals, leading to unstable data collection.

Second, we analyze the sensitivity of SynCoord to the teacher model capacity. Replacing the 235B teacher with a smaller 30B model only slightly reduces performance, indicating that the framework does not rely on a near-perfect oracle teacher. However, further reducing the teacher size to 8B substantially decreases both data collection accuracy and final success rate. This finding suggests that high-quality task-level coordination supervision benefits from sufficiently capable reasoning models.

Finally, we study the capacity requirement of the distilled intuitive agent. Although the 1.5B agent achieves relatively high single-agent behavioral accuracy, it completely fails in cooperative execution with a 0.00 success rate. We observe that the small model frequently falls into local action repetition and fails to maintain long-horizon coordination consistency. This result indicates that task-level coordination requires substantially higher representational capacity than local behavior imitation, while the 7B model provides an effective balance between inference efficiency and coordination capability.

Overall, these results demonstrate that SynCoord depends on the synergy between structured bootstrap supervision, high-capacity reasoning teachers, and sufficiently expressive distilled coordination policies.

In addition to performance, inference efficiency differs substantially across methods. The SynCoord distilled model achieves an average decision latency of 350 ms, compared to 15609 ms for Qwen3-235B-A22B (Thinking Agent). This reduction enables practical deployment in real-time multi-agent control loops, while preserving task-level coordination quality.

## 4.7. Analysis of Coordination Patterns

We analyze the coordination behaviors induced to better understand how task-level assignment systematically translates into effective multi-agent cooperation. Across the majority of layouts, we observe a consistent coordination pattern in which agents specialize in complementary tasks, i.e., while one agent performs ingredient collection, another concurrently handles dish collection. This form of implicit task parallelism emerges without explicit joint action modeling and remains stable even in visually and spatially highly complex environments.

Notably, this pattern generalizes across layouts with different bottlenecks and resource placements. As long as the task decomposition admits parallel execution (e.g., ingredient collection versus delivery), the distilled agent is able to assign non-conflicting tasks that reduce idle time and avoid agent interference. This suggests that task-level coordination alone is sufficient to capture a large portion of effective cooperative structure in Overcooked-style environments.

However, we also identify failure cases in more constrained layouts, as illustrated in Fig. 4c and Fig. 4d. In these scenarios, successful cooperation requires fine-grained temporal synchronization or dynamic task reassignment under tight spatial constraints. Since the distilled agent operates over predefined task abstractions and does not explicitly reason about future congestion or deadlock, agents may commit to locally valid but globally incompatible task assignments, leading to stalled progress or inefficient oscillations. In addition, Fig. 4d highlights a distinct failure mode in which effective cooperation relies on explicit ingredient passing between agents. This interaction is not encapsulated as a tool in our framework, and therefore cannot be discovered or exploited in the same way as in RL-based approaches that learn such low-level coordination behaviors through action-level optimization, as observed in prior work on Overcooked-AI.

These observations highlight a key trade-off of task-level coordination: while it enables robust and generalizable cooperation in most settings, its expressiveness is limited in layouts where successful execution critically depends on precise timing or adaptation beyond the predefined task set.

## 5. Discussion

**Summary.** This work introduces **SynCoord**, a framework that reframes generalizable multi-agent cooperation as a task-level coordination problem rather than a low-level joint control problem. By distilling high-capacity reasoning from "Thinking Agents" into a lightweight, real-time "Distilled Agent," we decouple deliberative task allocation from execution-time constraints. Experiments on the Overcooked-AI benchmark demonstrate that SynCoord achieves strong generalization across diverse layouts and unseen task com-

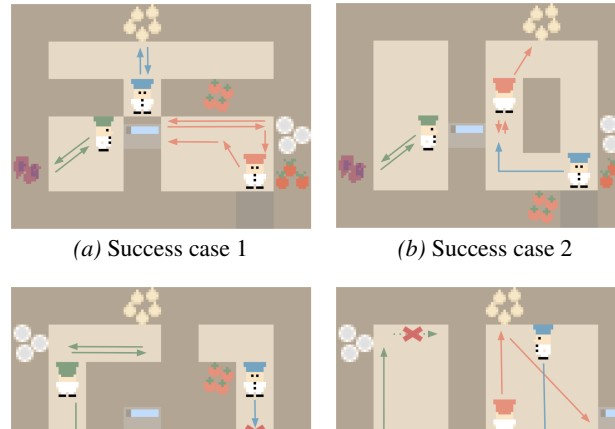

*(a)* Success case 1      *(b)* Success case 2

*(c)* Failure case 1      *(d)* Failure case 2

*Figure 4.* **Examples of different layouts and representative failure cases.** Success cases illustrate stable task parallelism enabled by task-level coordination, while failure cases highlight limitations in tight spatial constraints and interactions requiring fine-grained temporal synchronization.

positions while maintaining efficient execution. Furthermore, our distilled policy maintains high efficiency across unseen recipes and scales zero-shot to three-agent teams. Results suggest that task-level abstractions distilled from reasoning models provide an effective and computationally efficient approach for scalable multi-agent collaboration.

**Limitations and Future Work.** Despite its effectiveness, our approach relies on offline LLM supervision without direct environment interaction, which may limit adaptation under stochastic or rapidly changing environments. Second, the centralized paradigm assumes full observability, limiting its application to decentralized human-agent collaboration where intent inference is required. Finally, fixed task abstractions may restrict expressiveness in highly dynamic scenarios. Future work will explore online environment feedback, adaptive task abstractions, and explicit spatial reasoning to enhance coordination flexibility.

## Impact Statement

This work advances multi-agent generalization through LLM distillation, offering scalable coordination for robotics and logistics. We anticipate minimal negative societal impact as a simulation-based study; however, real-world deployment necessitates rigorous safety testing to mitigate risks under distribution shifts.

## Acknowledgments

This work is supported in part by the National Key R&D Program of China (2022ZD0114900).

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

# A. Preliminary Intuitive Agent Bootstrap

We collected 400 trajectories using a heuristic agent to finetune the Large Language Model, including completing the combination of 13 kinds of recipes under 40 different layouts. Different layouts are described in Fig. 5.

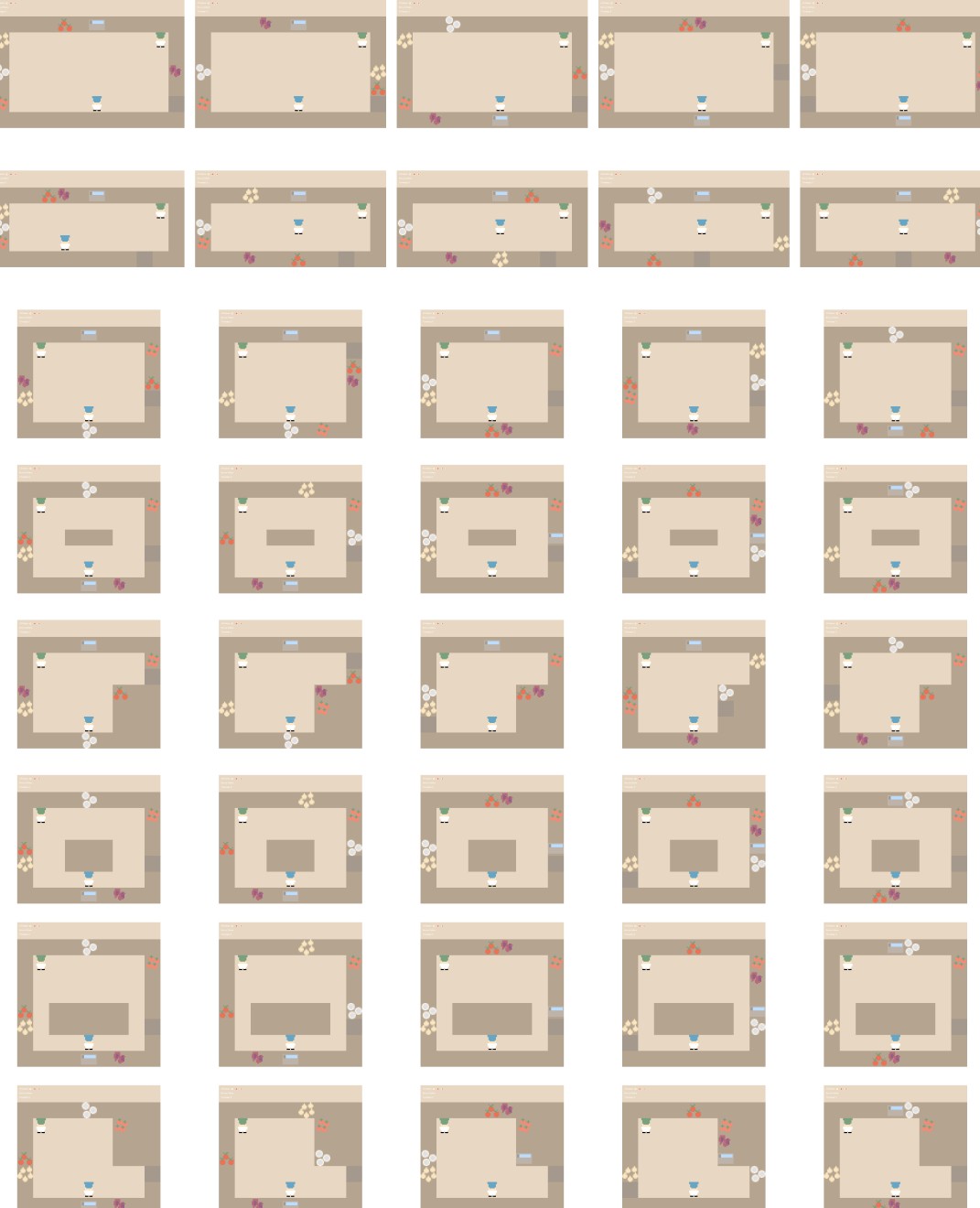

*Figure 5.* Intuitive agent training layouts

The recipes include: ('onion', 'onion', 'onion'), ('beef', 'beef'), ('onion', 'onion', 'tomato'), ('onion', 'onion'), ('onion', 'tomato', 'tomato'), ('tomato', 'tomato'), ('beef', 'onion', 'onion'), ('onion', 'tomato'), ('beef', 'onion'), ('tomato', 'tomato', 'tomato'), ('beef', 'tomato'), ('apple', 'apple'), ('apple', 'tomato'). We then test the model on 500 seen and unseen recipes and layouts.

The intuitive agent finetune data sample is shown below:

```
{
  "id": "77ef631d-42f5-4a6a-bd17-089f141f08ad",
  "conversations": [
    {
      "from": "human",
      "value": "You are a helpful assistant in kitchen. You have to: 1.pick up and deliver
          the ingredients to pot, one ingredient at a time, according to the receipes;2.
          start cooking;3.get the dish;4.go to the pot and get the food;5. go to the
          serving point. You must only generate actions involving objects listed after '
          You can access'. If an object is not listed there, you cannot interact with it.
          Do not output wait unless there is truly nothing else to do. You can access:
          dish, onion, pot, serving_loc. \nYour state: You are holding dish. The current
          order is: ['onion', 'onion']. Ingredients in the pot include onion, onion. Soup
          is cooking, need wait. Objects within reach include pot. "
    },
    {
      "from": "function_call",
      "value": "[{\"name\": \"wait\", \"arguments\": {}}]"
    }
  ],
  "tools": "[{\"name\": \"move_to\", \"description\": \"moving to certain position\", \"
      parameters\": {\"type\": \"object\", \"properties\": {\"target\": {\"type\": \"
      string\", \"enum\": [\"onion\", \"tomato\", \"apple\", \"beef\", \"dish\", \"pot\",
       \"serving_point\"], \"description\": \"target position, choose from 'onion', '
      tomato', 'apple', 'beef', 'pot', 'dish', 'serving point'\"} }, \"required\": [\"
      target\"]} }, {\"name\": \"pickup_item\", \"description\": \"pick up certain object
      \", \"parameters\": {\"type\": \"object\", \"properties\": {\"item_type\": {\"type
      \": \"string\", \"enum\": [\"onion\", \"tomato\", \"apple\", \"beef\"],
      \"description\": \"object type\"} }, \"required\": [\"item_type\"]} }, {\"name\":
      \"put_into_pot\", \"description\": \"put certain object into the pot, need holding
      the object after move to pot\", \"parameters\": {\"type\": \"object\", \"properties
      \": {} }, \"required\": []}, {\"name\": \"start_cooking\", \"description\": \"start
       cooking, need all objects in the pot\", \"parameters\": {\"type\": \"object\", \"
      properties\": {} }, \"required\": [] }, {\"name\": \"take_meal\", \"description\":
      \"take meal after moving to the pot when holding a dish, to get the cooked meal
      into the dish\", \"parameters\": {\"type\": \"object\", \"properties\": {} }, \"
      required\": []}, {\"name\": \"serve_meal\", \"description\": \"serve the cooked
      meal to the serving location, after move to the serving location and take meal into
       the dish\", \"parameters\": {\"type\": \"object\", \"properties\": {} }, \"
      required\": []}, {\"name\": \"wait\", \"description\": \"do nothing, just wait,
      when the objects are not ready or the agent is waiting for the other agent to
      finish\", \"parameters\": {\"type\": \"object\", \"properties\": {} }, \"required\":
       []}, {\"name\": \"step_back\", \"description\": \"step back, when the agent is in
      the way of the other agent\", \"parameters\": {\"type\": \"object\", \"properties\":
       {} }, \"required\": []} ]"
},
```

*Table 5.* Tools for function call

| Name | Description |
|------|-------------|
| move_to | moving to certain position |
| pickup_item | pick up certain object when it is nearby |
| put_into_pot | put certain object into the pot after move to pot, need holding the object |
| start_cooking | start cooking, need all objects in the pot, before move to the dish |
| take_meal | take meal after moving to the pot when holding a dish, to get the cooked meal into the dish |
| serve_meal | serve the cooked meal to the serving location, after move to the serving location and take meal into the dish |
| wait | do nothing, just wait, when the objects are not ready or the agent is waiting for the other agent to finish |
| step_back | step back, when the agent is in the way of the other agent |

As shown in Table 5, we designed several tools and they remain the same throughout the process.

```
OVERCOOKED_TOOLS = [
    {
        "type": "function",
        "function": {
            "name": "move_to",
            "description": "moving to certain position",
            "parameters": {
                "type": "object",
                "properties": {
                    "target": {
                        "type": "string",
                        "enum": ["onion", "tomato", "apple", "beef", "dish", "pot", "serving_loc
                            "],
                        "description": "target position, choose from 'onion', 'tomato', 'apple',
                            'beef', 'pot', 'dish', 'serving_loc'"
                    }
                },
                "required": ["target"]
            }
        }
    },
    {
        "type": "function",
        "function": {
            "name": "pickup_item",
            "description": "pick up certain object when it is nearby",
            "parameters": {
                "type": "object",
                "properties": {
                    "item_type": {
                        "type": "string",
                        "enum": ["onion", "tomato", "apple", "beef", "dish"],
                        "description": "object type"
                    }
                },
                "required": ["item_type"]
            }
        }
    },
    {
        "type": "function",
        "function": {
            "name": "put_into_pot",
            "description": "put certain object into the pot after move to pot, need holding
```

```
          the object",
        "parameters": {
          "type": "object",
          "properties": {}
        },
        "required": []
    }
  },
  {
    "type": "function",
    "function": {
      "name": "start_cooking",
      "description": "start cooking, need all objects in the pot, before move to the
          dish",
      "parameters": {
        "type": "object",
        "properties": {}
      },
      "required": []
    }
  },
  {
    "type": "function",
    "function": {
      "name": "take_meal",
      "description": "take meal after moving to the pot when holding a dish, to get the
          cooked meal into the dish",
      "parameters": {
        "type": "object",
        "properties": {}
      },
      "required": []
    }
  },
  {
    "type": "function",
    "function": {
      "name": "serve_meal",
      "description": "serve the cooked meal to the serving location, after move to the
          serving location and take meal into the dish.",
      "parameters": {
        "type": "object",
        "properties": {}
      },
      "required": []
    }
  },
  {
    "type": "function",
    "function": {
      "name": "wait",
      "description": "do nothing, just wait, when the objects are not ready or the
          agent is waiting for the other agent to finish",
      "parameters": {
        "type": "object",
        "properties": {}
      },
      "required": []
    }
  },
  {
    "type": "function",
    "function": {
      "name": "step_back",
      "description": "step back, when the agent is in the way of the other agent",
```

```
      "parameters": {
        "type": "object",
        "properties": {}
      },
      "required": []
    },
  },
]
```

## B. Distilled Agent Training

We used the same recipes as those employed in training the Intuitive Agent, including all 13 recipes. The distilled agent finetune data sample is shown below:

```
{
  "id": "61b43620-b2ea-482e-8480-63fac29546c8",
  "conversations": [
    {
      "from": "system",
      "value": "\nYou are an Overcooked game task allocator. Based on the current state,
          select the best task for the agent.\n\nCurrent state:\n- Agent agent_1: holding
          nothing, can access ['beef', 'wall', 'onion', 'pot']\n- Another agent: holding
          tomato, status busy, action queue [[-1, 0], [0, 1], [0, 1]]\n- Current order: ['
          tomato', 'tomato']\n- Task queue: ['pickup_tomato', 'deliver_tomato', '
          start_cooking', 'get_dish', 'get_soup', 'serve_dish']\n- Pot state: {'has_soup':
           False, 'soup_ready': False, 'ingredients': [], 'ingredients_match_order': False
          }\n- Completed tasks: ['pickup_tomato']\n- Whether this agent has completed
          get_soup before: False\n\nTask execution conditions (must be strictly checked):\
          n1. pickup_X task:\n - The agent must be empty-handed (holding nothing)\n - The
          agent must be able to access object X\n\n2. deliver_X task:\n - The agent must
          be holding item X\n - The agent must be able to access the pot\n - X must be in
          the agent's accessible object list\n\n3. start_cooking task:\n - The agent must
          be empty-handed\n - The agent must be able to access the pot\n - The pot must
          contain soup (has_soup is true)\n - The ingredients in the pot must exactly
          match the order (ingredients_match_order is true)\n\n4. get_dish task:\n - The
          agent must be empty-handed\n - The agent must be able to access a dish\n\n5.
          get_soup task:\n - The agent must be holding a dish\n - The agent must be able
          to access the pot\n - The soup must already be ready (soup_ready is true)\n\n6.
          serve_dish task:\n - The agent must be holding soup\n - This agent must have
          completed the get_soup task before\n\nPlease strictly check the feasibility of
          each task according to the above conditions and select the first task that
          satisfies all conditions. If no task is feasible, return -1.\n"
    },
    {
      "from": "human",
      "value": "Please select the best task for agent_1"
    },
    {
      "from": "function_call",
      "value": "[{\"name\": \"select_task\", \"arguments\": {\"selected_task_index\": -1,
          \"reason\": \"No feasible task can be executed\"}}]"
    }
  ],
  "tools": "[{\"type\": \"function\", \"function\": {\"name\": \"select_task\", \"
      description\": \"Analyze the task queue and select the optimal task\", \"parameters
      \": {\"type\": \"object\", \"properties\": {\"selected_task_index\": {\"type\": \"
      integer\", \"description\": \"Index of the selected task in the queue; return -1 if
       no feasible task exists\"}, \"reason\": {\"type\": \"string\", \"description\": \"
      Reason for selecting this task\"}}, \"required\": [\"selected_task_index\", \"
      reason\"]}}}]"
},
```

```
tools = [
    {
        "type": "function",
        "function": {
            "name": "select_task",
            "description": "Analyze the task queue and select the optimal task",
            "parameters": {
                "type": "object",
                "properties": {
                    "selected_task_index": {
                        "type": "integer",
                        "description": "Index of the selected task in the task queue; return -1
                            if no feasible task exists"
                    },
                    "reason": {
                        "type": "string",
                        "description": "Reason for selecting this task"
                    }
                },
                "required": ["selected_task_index", "reason"]
            }
        }
    }
]
```

System prompt:

```
system_prompt = f"""
You are an Overcooked game task allocator. Based on the current state, select the best
    task for the agent.

Current state:
- Agent {serializable_state['agent_id']}: holding {serializable_state['agent_held'] or '
    nothing'}, can access {serializable_state['agent_accessible']}
- Another agent: holding {serializable_state['other_agent_held'] or 'nothing'}, status {
    serializable_state['other_agent_status']}, action queue {serializable_state['
    other_agent_actions']}
- Current order: {serializable_state['current_order']}
- Task queue: {serializable_state['task_queue']}
- Pot state: {serializable_state['pot_status']}
- Completed tasks: {serializable_state['completed_tasks']}
- Whether this agent has completed get_soup before: {serializable_state['
    get_soup_completed_by_agent']}

Task execution conditions (must be strictly checked):
1. pickup_X task:
   - The agent must be empty-handed (holding nothing)
   - The agent must be able to access object X

2. deliver_X task:
   - The agent must be holding item X
   - The agent must be able to access the pot
   - X must be in the agent's accessible object list

3. start_cooking task:
   - The agent must be empty-handed
   - The agent must be able to access the pot
   - The pot must contain soup (has_soup is true)
   - The ingredients in the pot must exactly match the order (ingredients_match_order is
       true)

4. get_dish task:
   - The agent must be empty-handed
```

```
  - The agent must be able to access a dish

5. get_soup task:
  - The agent must be holding a dish
  - The agent must be able to access the pot
  - The soup must already be ready (soup_ready is true)

6. serve_dish task:
  - The agent must be holding soup
  - This agent must have completed the get_soup task before

Please strictly check the feasibility of each task according to the above conditions and
    select the first task that satisfies all conditions. If no task is feasible, return -1.

Single-agent prompt: {getattr(state_info, 'single_agent_prompt', '')}
"""
```

## C. Experiment results

**Recipe selection** Except for the original ingredients "onion" and "tomato" in Overcooked-AI(Carroll et al., 2019), we introduce "beef" and "apple" to increase task variety and evaluate the agents' ability to handle unseen recipes. Each recipe consists of 2–2-3 ingredients.

**Action space** To better align with real-world action semantics, we designed a higher-level action space (move_to, pickup_item, start_cooking, etc.) for the LLM agent, instead of relying on the single overloaded interact action in the standard Overcooked environment. This makes the agent's reasoning more interpretable and generalizable, but also increases the planning overhead, leading to slower execution compared to RL agents operating in the simplified environment.

**Baselines** We evaluated success rate and average success timesteps of our approach against three multi-agent reinforcement learning baselines including MAPPO, RMAPPO and IPPO.

- **MAPPO**: parameter-shared PPO with a centralized critic (centralized value for training) as described in previous MARL work;

- **R-MAPPO**: recurrent MAPPO (MAPPO with a recurrent policy/encoder) to enable partial-observation memory.

- **IPPO** (Independent PPO): per-agent PPO training with no centralized critic;

All baselines use the same input preprocessing and encoder architecture (except for the RNN module in R-MAPPO) to ensure fairness: differences in results reflect algorithmic training dynamics rather than representation differences.

### C.1. Intuitive agent Results

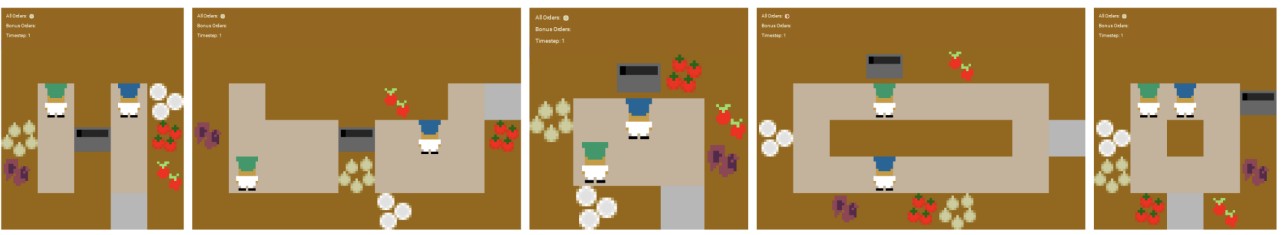

*Figure 6.* From left to right: *Forced Coordination* representing Disjoint Layouts, *Asymmetic Advantages* representing Overlap Layouts, *Cramped Room* representing Open Layouts, *Counter Circuit* and *Coordination Ring* representing Ring Layouts.

To evaluate the ability of the Intuitive Agent, we tested it on the training set containing 400 samples, as well as unseen layouts and unseen recipes.

| Split | Correct | Total | Accuracy (%) |
|---|---|---|---|
| Seen | 382 | 400 | 95.5 |
| Unseen | 78 | 100 | 78.0 |

*Table 6.* Performance of the Intuitive Agent on seen and unseen recipes.

## C.2. Comparison with RL-based Coordination

*Table 7.* Performance comparison under different training distributions. SynCoord is trained on 5 layouts $\times$ 13 scenes, while RL baselines are trained on a single layout. All are tested on their trained layouts.

| | Disjoint | | | Overlap | | | Open | | |
|---|---|---|---|---|---|---|---|---|---|
| Method | SR | Steps | Reward | SR | Steps | Reward | SR | Steps | Reward |
| MAPPO($1\times1$) | 1.00 | 37.08 | 38.23 | 1.00 | 47.23 | 41.77 | 1.00 | 43.00 | 39.31 |
| MAPPO($1\times13$) | 0.15 | 47.50 | 45.50 | 0.31 | 50.00 | 42.50 | 0.15 | 48.50 | 44.00 |
| RMAPPO($1\times1$) | 1.00 | 36.46 | 37.85 | 1.00 | 42.00 | 38.77 | 1.00 | 37.00 | 35.15 |
| RMAPPO($1\times13$) | 0.08 | 28.00 | 34.00 | 0.38 | 40.80 | 36.40 | 0.08 | 42.00 | 37.00 |
| IPPO($1\times1$) | 1.00 | 34.08 | 36.54 | 1.00 | 40.54 | 37.69 | 1.00 | 40.31 | 38.08 |
| IPPO($1\times13$) | 0.08 | 37.00 | 37.00 | 0.38 | 73.40 | 48.78 | 0.15 | 58.50 | 45.50 |
| SynCoord | 1.00 | 37.54 | 35.15 | 1.00 | 39.86 | 35.15 | 0.92 | 42.50 | 35.04 |

*Table 8.* Performance comparison under different training distributions. SynCoord is trained on 5 layouts $\times$ 13 scenes, while RL baselines are trained on a single layout. All are tested on their trained layouts.

| | Ring1 | | | Ring2 | | | all | | |
|---|---|---|---|---|---|---|---|---|---|
| Method | SR | Steps | Reward | SR | Steps | Reward | SR | Steps | Reward |
| SynCoord | 1.00 | 49.35 | 35.15 | 1.00 | 64.97 | 35.15 | 0.98 | 46.85 | 35.13 |

