# OpenReview forum: "Distilling Task-Level Coordination Policies for Generalizable Multi-Agent Cooperation"
_ICML.cc/2026/Conference — ICML 2026 regular_

### Official Review · Reviewer_WKdR · 2026-02-25

**Soundness:** 2
**Presentation:** 3
**Significance:** 3
**Originality:** 3
**Overall Recommendation:** 4
**Confidence:** 4

**Summary:**

1. This paper presents SynCoord, a two-stage framework in which a high-capacity thinking agent generates task-level coordination trajectories, which are subsequently distilled into lightweight, execution-time policies designed to operate under partial observability. By decoupling deliberative reasoning from real-time control, the approach eliminates the need for direct LLM inference during deployment.
2. The method is primarily evaluated on the Overcooked-AI benchmark, showing high success rates, superior execution efficiency relative to RL baselines, and robust generalization across varied layouts, novel recipes, and different team sizes.

**Compliance With Llm Reviewing Policy:**

Affirmed.

**Final Justification:**

I appreciate the authors' efforts on addressing my concerns. The only remaining issue is that they cannot provide the supplementary experimental results within the rebuttal period. I believe my current evaluation is fair enough, so I decide to maintain my score.

**Key Questions For Authors:**

1. **Reliability of LLMs as teachers.** The method assumes that large LLMs can effectively generate optimal coordination strategies. However, how does the framework handle situations in which the tasks are particularly complex, long-horizon, or outside the LLMs' core reasoning capabilities, leading the thinking agent to generate suboptimal or inconsistent coordination?
2. **Generalization beyond Overcooked-AI.** While the paper claims broad generalizability, the experimental validation is limited to Overcooked-AI. Could the approach be evaluated on additional established multi-agent cooperation benchmarks, such as Hanabi, to better demonstrate its generalization capabilities?
3. **Choice of teacher model.** Though Qwen3-235B-A22B is a strong model, why not use state-of-the-art models such as GPT-5 or Gemini-3 as teachers? How sensitive is the distilled policy to the teacher model quality?

**Limitations:**

The limitations and potential negative societal impact have been discussed in this paper.

**Strengths And Weaknesses:**

### Strengths

1. **Simple and effective framework.** The proposed distillation pipeline is conceptually clean and technically straightforward. The separation between a deliberative teacher model and a lightweight execution model is well-motivated and practically appealing.
2. **Valuable methodology for lightweight coordination training.** The framework provides a practical methodology for training lightweight agents to learn coordination on downstream tasks.
3. **Strong empirical results within the chosen benchmark.** On Overcooked-AI, SynCoord achieves competitive or superior performance compared to RL baselines and training-free LLM baselines.

### Weaknesses

1. **Assumption of reliable LLM teachers.** The approach rests on the crucial assumption that large LLMs can reliably generate effective coordination strategies. In practice, real-world cooperative tasks may be far more intricate. If the teacher model's coordination is unreliable or flawed, it risks undermining the entire distillation process by propagating suboptimal behaviors.
2. **Narrow experimental scope.** The empirical validation is restricted solely to the Overcooked-AI environment, despite the paper's claims of broad generalizability. The extent to which the method transfers to other cooperative multi-agent domains remains untested, thereby limiting the strength of claims regarding its general coordination capabilities.
3. **Implicit acquisition of coordination skills.** While the thinking agent produces explicit coordination plans, the distilled agent only observes the resulting actions. Thus, the acquisition of coordination skills by the execution policy remains implicit, potentially constraining the fidelity with which complex coordination behaviors are learned.

---

### Official Review · Reviewer_o2aU · 2026-02-28

**Soundness:** 3
**Presentation:** 3
**Significance:** 2
**Originality:** 3
**Overall Recommendation:** 4
**Confidence:** 4

**Summary:**

The paper proposes using large language models (LLMs) to coordinate multiple agents in multi-agent environments. It contrasts this approach with reinforcement learning (RL)-based coordination methods, arguing that RL typically requires extensive scenario-specific training and often struggles to generalize across tasks or environments. In contrast, LLM-based coordination does not require task-specific training, enabling more flexible deployment. However, reasoning over temporal constraints and long-horizon dependencies remains a challenge for LLMs.

The proposed framework adopts a hierarchical structure: lightweight LLM-based agents make local decisions, while higher-level agents coordinate global objectives and inter-agent interactions. The paper investigates whether such a layered LLM-driven approach can achieve effective multi-agent coordination without the heavy training overhead of RL.

**Compliance With Llm Reviewing Policy:**

Affirmed.

**Key Questions For Authors:**

The role of the LLMs in the overall framework is not entirely clear to me. Could the authors specify where exactly the LLMs are incorporated into the algorithm, what inputs they receive, and how their outputs are utilized during training and inference?

**Limitations:**

Yes: The approach relies on offline LLM supervision without direct interaction with the environment, which may lead to suboptimal performance in stochastic settings. It also assumes centralized, fully observable coordination, limiting applicability to decentralized or human-agent scenarios. Additionally, fixed task abstractions may reduce flexibility in highly dynamic environments. Future work aims to address these limitations through online grounding and adaptive task representations.

**Strengths And Weaknesses:**

The related work is competent.
The problem descriptions i also clear (and standard). A collection of n agents, each doing a vector \tilde{a} of actions (one for each): (a1,...,an). These is a continuing task with discounted rewards over the vector of states (one for each state).

The thinking agent takes action vectors, explanations and coordination rationale. The intuitive agent trains how to follow the high-level trajectories generated by the thinking agent.

 I don't find the idea particularly compelling. These kind of two-stage architectures are frequently proposed: for instance, H-DQN follows a similar idea: one layer produces high level tasks and the other learns how to execute it. This is a very similar idea here. While here it is on muli-agents; in fact, the multi-agents are treated as basically a single agent composite of multiple agents \vector(a) = (a1,...an).

 Separate from this, it isn't entirely clear to me of the roles of LLMs here. I appreciate the algorithm but it is not clear from there what the LLMs are doing.

Their system outperforms RL agents. It also outperform LLM-only agents, revealing that the LLM-only struggles with temporal consistency. It also demonstrates some level of generalizability. The system is robust to increase in agents. The paper also presents an ablation study showing that learned coordination is necessary, distillation effectively translate high-level tasks into effective coordination and guided supervision reduces failures. It shows that agents can coordinate.


Overall, the empirical section supports the claim that the agents can learn to coordinate effectively under the proposed framework. However, the conceptual contribution appears incremental relative to established two-tier  approaches such as H-DQN, and the what precisely the LLMs are doing in the algorithm is not clear.

---

### Official Review · Reviewer_aA6o · 2026-03-10

**Soundness:** 2
**Presentation:** 2
**Significance:** 2
**Originality:** 3
**Overall Recommendation:** 3
**Confidence:** 3

**Summary:**

The paper shows that distillation from a strong teacher policy (Qwen3-235B-Instruct) on Overcooked tasks leads to better multi-agent performance than using strong RL baselines. They illustrate a three-stage pipeline (1) bootstrapping from single agent trajectories on the same dataset, (2) dataset generation using a strong teacher model, and (3) behavior cloning to the smaller model. The large SoTA models have already achieved near-optimal performance, and the method proposed here, SynCoord, also achieves a success rate of 0.97. However, the method is able to achieve this with, on average, 7 fewer timesteps.

**Compliance With Llm Reviewing Policy:**

Affirmed.

**Final Justification:**

The rebuttal has partially addressed my concerns regarding the ablations and the teacher model, but it does not make it clear whether simpler distillation techniques could perform similarly well. Therefore, I will increase my score from reject to weak reject. I will not raise my score further than this, but I am still uncertain of whether the full complexity of the method is necessary and how this affects final performance.

**Key Questions For Authors:**

1. What is the most important part of the SynCoord algorithm? The bootstrap, the intuitive agent, or the strong teacher model and thinking agent?
2. Do you think that SynCoord would work well in scenarios where the teacher model is not well-suited? Can we combine this sort of distillation with reinforcement learning in the future?
3. In SynCoord, we only distill an agent as the instruction follower; could we also distill an agent for better high-level planning over coordination and global state information?

**Limitations:**

Yes

**Strengths And Weaknesses:**

**Strengths**

1. The SynCoord method clearly results in a strong empirical performance, with a 0.97 success rate with fewer timesteps than other previous methods.
2. The paper presents failure modes and how distilled models fail in unique ways, e.g., selecting tasks that are locally optimal but may lead to deadlock or duplication of tasks.
3. The technique proposed by SynCoord is complex and original.

**Weaknesses**

1. While the paper presents several ablations, it does not present a clear ablation of which elements of their pipeline are the most critical for improving performance. Table 3 provides some ablations, but more ablations showing the effect of each component of the pipeline would be helpful, e.g., removing the bootstrap, changing the teacher model, or changing the intuitive agent.
2. While the work shows strong empirical results, the teacher models have already achieved near-optimal performance. Thus, it does not seem fair to directly compare this to reinforcement learning, as it is clear that distillation from a near-optimal teacher model would outperform reinforcement learning from scratch.
3. The idea of distillation on complex tasks is not new, though the exact mechanism of SynCoord is new. Similar to point (1), how do we know that a simpler technique would not perform similarly well?

---

### Official Review · Reviewer_6iqk · 2026-03-13

**Soundness:** 3
**Presentation:** 3
**Significance:** 2
**Originality:** 2
**Overall Recommendation:** 3
**Confidence:** 3

**Summary:**

This paper proposes SynCoord (Synthetic Coordination Distillation), a self-supervised framework that distills high-level coordination behaviors from large language models with advanced reasoning ability into much more lightweight ones. Experiments on Overcooked-AI show that the distilled policies achieve performance comparable to RL-based methods while reducing inference cost and improving generalization across team sizes.

**Compliance With Llm Reviewing Policy:**

Affirmed.

**Final Justification:**

If the authors can do more experiments on a reasonable domain, I would like to raise my score to 4. The authors said "We will include these results in the next few days before the end of the rebuttal phase." but somehow I can see nothing. I also can't see rebuttal for other reviewers, their comments, etc.

**Key Questions For Authors:**

- Do you have any results on other domains that require multi-agent collaboration?

**Limitations:**

Yes

**Strengths And Weaknesses:**

Strengths
- The proposed idea shows some interesting performance, by trying to distill the high level decisions of a large reasoning model to much smaller models to give its some strategic level knowledge, the model demonstrates some generalization ability.
- The paper is well-written and easy to follow.

Weaknesses
- The experiments are conducted only on overcooked environment, which seems to be inefficient, as most of the times people show experiment results on multiple domains for reinforcement learning or foundation model agent methods, unless the domain is very challenging or cover a lot of different aspects already. The current environment does not look that complicated, especially that they use a structured tool interface so that the model still generate text-based high-level actions instead of more fine-grained low level actions.
- The setting is still a bit confusing to me. In terms of some of the presented failure cases in the experiment, the model fails because it does not learn about cooperation related dynamics as the initial environment dynamics related data are collected from a single agent scenarios. However, it is not surprising that the reasoning model's imagination alone cannot capture that at all. If you do not want to analyze cooperation dynamics, and only care about whether generating a high level collaboration plan is feasible, then simply make the map larger so that spatial crowding happens less seem to make more sense for cleaner and more comprehensive analysis and strength the claim. However, at the same time, these experiments demonstrate that the large language model's imagination works this self-supervised way partially because the cooperation scenario is too simple.

---

### Decision · Program_Chairs · 2026-04-30

**Decision:**

Accept (regular)

**Comment:**

This paper proposes SynCoord, a framework for distilling coordination knowledge from LLM-based planners into smaller, deployable multi-agent RL policies. The core idea of using LLMs as a coordination prior for MARL training is interesting and addresses a real deployment gap. Reviewers generally found the problem well-motivated and the approach sensible.

However, I share the concern raised by Reviewer 6iqk that the experimental evaluation is limited to a single domain (Overcooked), and the newly added LBF environment is of a similar flavor. To convincingly demonstrate the generality of this framework, the authors should evaluate on more complex and structurally diverse multi-agent environments. Additionally, I agree with the authors that distilling from a near-optimal high-level planner is not a directly comparable baseline to RL-based coordination learning. That said, given that the method assumes access to global information during distillation, the authors should compare against simpler distillation or imitation-based methods to better isolate what SynCoord's LLM-based coordination specifically contributes.

I also note that there was a procedural issue during the rebuttal period: the authors' response was posted as a public comment rather than an official rebuttal, which Reviewer 6iqk did not see before finalizing their assessment. While unfortunate, the substantive concerns about evaluation breadth and baselines remain valid regardless.